# Plant Extract-Derived Carbon Dots as Cosmetic Ingredients

**DOI:** 10.3390/nano13192654

**Published:** 2023-09-27

**Authors:** Le Thi Nhu Ngoc, Ju-Young Moon, Young-Chul Lee

**Affiliations:** 1Department of Nano Science and Technology Convergence, Gachon University, 1342 Seongnam-Daero, Sujeong-gu, Seongnam-si 13120, Gyeonggi-do, Republic of Korea; nhungocle92@gmail.com; 2Major in Beauty Convergence, Kwangwoon University, 20 Kwangwoon-ro, Nowon-gu, Seoul 01897, Republic of Korea; 3Department of BioNano Technology, Gachon University, 1342 Seongnam-daero, Sujeong-gu, Seongnam-si 13120, Gyeonggi-do, Republic of Korea

**Keywords:** carbon dots, plant extracts, skin-aging, anti-inflammation, UV protection

## Abstract

Plant extract-derived carbon dots (C-dots) have emerged as promising components for sustainability and natural inspiration to meet consumer demands. This review comprehensively explores the potential applications of C-dots derived from plant extracts in cosmetics. This paper discusses the synthesis methodologies for the generation of C-dots from plant precursors, including pyrolysis carbonization, chemical oxidation, hydrothermal, microwave-assisted, and ultrasonic methods. Plant extract-derived C-dots offer distinct advantages over conventional synthetic materials by taking advantage of the inherent properties of plants, such as antioxidant, anti-inflammatory, and UV protective properties. These outstanding properties are critical for novel cosmetic applications such as for controlling skin aging, the treatment of inflammatory skin conditions, and sunscreen. In conclusion, plant extract-derived C-dots combine cutting-edge nanotechnology and sustainable cosmetic innovation, presenting an opportunity to revolutionize the industry by offering enhanced properties while embracing eco-friendly practices.

## 1. Introduction

Carbon dots (C-dots) are a new type of fluorescent carbon materials with a size of less than 10 nm [1,2]. C-dots consist of a large amount of carbon and a small amount of other elements (H, N, and O) presenting on the surface. The structure of C-dots is characterized by a graphite core (sp^2^ hybridized carbon) and an amorphous carbon shell [2,3]. C-dots are a potential material alternative to quantum dots generated from metal due to their high water solubility, photoluminescence properties, biocompatibility, low toxicity, and high chemical stability [3]. These properties are mainly attributed to their nanosize and surface functional groups, which allow tailored interactions with biomolecules and other materials [4]. Thus, C-dots exhibit high potential for applications in many fields, such as catalysis [5], bioimaging [6], energy conversion [7], clinical diagnosis, controlled drug delivery [8], and cosmetics [9].

Recently, the field of cosmetics has experienced an increasing demand for innovative, sustainable, and nature-inspired ingredients that consider aesthetic and environmental factors. The exploration of sustainable alternatives to traditional cosmetic ingredients has led to the innovation of plant extract-derived compounds. Plant extracts have long been used for their therapeutic and cosmetic benefits [10,11]. The incorporation of plant extract-derived C-dots presents a novel avenue for the intersection of nanotechnology and natural ingredients. By harnessing the innate properties of plant-based precursors, C-dots can inherit desirable attributes, such as antioxidant properties, skin-soothing effects, and compatibility with various formulations, thereby accentuating their appeal in cosmetic applications [9,10,11]. Therefore, plant extract-derived C-dots should be considered novel ingredients owing to their inherent biodegradability, renewability, and potential to align with the increasing trend of green chemistry in the cosmetic industry.

This study investigated plant extract-derived C-dots and their increasing role in cosmetic applications. By harnessing the inherent properties of these nanomaterials, cosmetics can unlock innovative solutions that meet consumer demands and align with the contemporary values of sustainability and responsible product development. This review focuses on the synthesis methods for the production of C-dots from plant extracts. In particular, we explore how these C-dots can enhance cosmetic formulations: from pigmentation modulation, anti-inflammation, and skin moisturization to sunscreen protection. This paper provides an insightful overview of current research in this exciting field, providing insight on the beneficial effects of these C-dots in revolutionizing cosmetic formulations while aligning with eco-conscious consumer preferences.

## 2. Methods for the Synthesis of C-Dots from Plant Extracts

There are several approaches for synthesizing C-dots from carbon sources (e.g., carbon fiber, carbon nanotubes, graphite, graphene oxide, and organic molecules). These include pyrolysis, chemical oxidation, hydrothermal, microwave-assisted, and ultrasonic techniques [3,12].

### 2.1. Pyrolysis Carbonization Method

Pyrolysis carbonization is a basic approach for preparing C-dots. It has may advantages, including low toxicity, convenience, and simplicity [13]. The C-dot preparation can often be completed in one step without excessive equipment or complicated processes (Figure 1a) [13]. This method involves heating organic materials in the absence of oxygen to convert them into carbon-rich materials. This process breaks down the complex organic molecules in the extract, leaving carbon-rich fragments, which form the C-dots [13,14]. After the carbonization process, the resulting C-dots may require additional processing steps to enhance their properties, including washing, filtration, or further chemical processes to modify the surface chemistry of the C-dots for specific applications [13,14]. Centeno et al. (2021) successfully prepared C-dots from *Opuntia* sp. extract via microwave pyrolysis [14]. The *Opuntia* sp. extract was mixed with 2.5 mL of ultrapure water and 10 mL ethylene glycol, and then the mixture was magnetically stirred for 24 h. After 24 h, the reaction solution was pyrolyzed using a microwave oven (600 W for 4 min) to produce C-dots. The pyrolyzed solution was then centrifuged, washed, and dried under ambient conditions to obtain the final C-dot product [14].

### 2.2. Hydrothermal Method

Hydrothermal synthesis is the most popular approach for synthesizing C-dots because it is easy to perform and can produce homogenous particle sizes with high quantum yields [12,15]. This technique involves the controlled reaction of carbon-rich precursors under high-temperature (150–300 °C) and high-pressure conditions in a water-based solution. This method forms C-dots with tunable properties, such as surface chemistry, size, and luminescence (Figure 1b). Thus, the technique offers versatility and reproducibility in generating C-dots with the desired characteristics. For instance, Prathap et al. (2023) utilized a hydrothermal method to produce highly fluorescent plant extract-derived C-dots from *Prosopis juliflora* (*P. juliflora*) leaf extract [16]. The *P. juliflora*-derived C-dots exhibited emission/excitation peaks at wavelength of 540 nm/480 nm, and the absolute photoluminescence quantum yield was 7.88%, suggesting high photoluminescence activity [16]. In addition, C-dots substantially inhibited the growth of *Staphylococcus aureus* and were powerful components for bioimaging in *Caenorhabditis elegans*, exhibiting potential pharmaceutical applications [16]. Rammanarayanan et al. (2020) successfully synthesized C-dots from *Guava* leaf extract via a hydrothermal method [17]. The *Guava* leaf extract was heated at 160 °C for 1 h on a Teflon-lined stainless-steel autoclave. After cooling the autoclave, dark brownish C-dot solution was obtained, which exhibited good stability and green emission [17].

### 2.3. Microwave-Assisted Method

Microwave-assisted techniques can carbonize or pyrolyze carbon-rich precursors under microwave irradiation with mechanisms for transferring energy from microwaves to the heated substance [10]. In addition, microwave irradiation has penetrating properties that allow the quick and uniform heating of the reaction solution [4,10]. Compared with traditional time-consuming hydrothermal procedures, microwave-assisted processes can facilitate quick and easy one-pot synthesis. They are efficient, have low solvent consumption and high reproducibility, and produce residual bioactive compounds [7,8]. Despite the high energy demand, this simple technique can synthesize particles with controllable size and high quantum yields [7,8]. However, this method is limited by uncontrollable reaction conditions [9]. To generate plant extract-derived C-dots, plant materials are first extracted with a suitable solvent (typically water or ethanol), and the plant extracts are then mixed with precursor carbon sources (e.g., carbohydrates such as glucose) in an appropriate ratio (Figure 1c) [7,9]. Microwave-absorbing additives (e.g., polyethylene glycol) can be added to the mixture to enhance the absorption of microwave energy and promote uniform heating [7,9]. The mixtures are then heated in a microwave to promote carbonization and form carbon dots [7,9]. Sharma et al. (2022) synthesized biocompatible fluorescent C-dots from a medicinal plant *Calotropis gigantea* [18]. The C-dots had a diameter in the range of 2.7–10.4 nm with a 4.24% fluorescence quantum yield, suggesting that they can be a potential probe for optical and bio-imaging of plant cells, fungi, and bacterial [18]. Alsalem et al. (2023) reported *Vachellia nilotica*-derived C-dots using a microwave-assisted technique (900 W) for 30–45 min that exhibited green emission at λ_em_ = 480 nm/λ_ex_ = 386.5 nm and red fluorescence emission at λ_em_ = 673.9 nm/λ_ex_ = 410.5 nm [19].

### 2.4. Chemical Oxidation Synthesis

Chemical oxidation methods involve the oxidative treatment of carbon precursors using strong oxidants (sulfuric acid (H_2_SO_4_) or nitric acid (HNO_3_)) to produce C-dots [4]. First, the plant extract is added to an oxidizing agent (HNO_3_, H_2_SO_4_, or KMnO_4_) under suitable conditions (concentration, temperature, and reaction time) [4]. The reaction mixture is then heated using a microwave reactor or conventional heater to control the reaction kinetics and promote C-dot formation [4]. The mixture is then cooled and purified to obtain the final C-dot solution (Figure 1d). Hu et al. (2020) utilized green tea leaves to produce C-dots by combining oxidation with concentrated H_2_SO_4_ and pyrolysis at high temperatures [20]. First, the tea leaves were dried at 80 °C and ground into powder; the powder was then calcinated at 350 °C for 2 h under atmospheric conditions [20]. Subsequently, the resulting residue was carbonized with concentrated H_2_SO_4_ for 20 h, and C-dots were obtained after neutralization and centrifugation. The generated C-dots showed a great quantum yield (14.8%) and were stable in acidic environments and under ionic strength and ultraviolet radiation [20]. The desired properties of C-dots can be achieved by optimizing the synthesis parameters (precursor choice, oxidation agents, reaction conditions, and purification methods). The chemical oxidation approach offers numerous advantages, such as simplicity, versatility, scalability, and high fluorescence materials. However, this method often leads to the release of toxic gases, leading to increased cost and resulting in more cost to eliminate the excess acid or salt, limiting the application of this technique. To address this drawback, hydroxyl peroxide (H_2_O_2_) is used as an oxidant instead of an oxidizing acid to prepare C-dots with high photocatalytic activity [21]. Although the C-dots generated through the chemical oxidation exhibit high photoluminescence, the chemical toxicity should be paid attention to.

### 2.5. Ultrasonic Method

Ultrasonic synthesis is achieved via the thermal effects of ultra-high-frequency vibrations and cavitation [4,8]. This process involves breaking down the plant material using ultrasonic waves to release various biomolecules, which can then undergo carbonization to form C-dots [8]. Plant extracts are subjected to ultrasonic waves that break down the plant cell walls and facilitate the release of bioactive molecules [2,4]. After concentrating the plant extract solution, the condensed extract is heated in an oven or microwave [4]. Carbonization leads to the formation of C-dots (Figure 1d) [8]. For example, Zaib et al. (2021) prepared C-dots from *Polyalthia longifolia* leaf powder through subjecting it to ultra-high-frequency vibrations for 1 h [22]. Ultrasonic techniques are often used to synthesize C-dots from plant extracts owing to their efficiency, simplicity, eco-friendliness, small size, and uniform surface morphology materials [4,8].

**Figure 1 nanomaterials-13-02654-f001:**
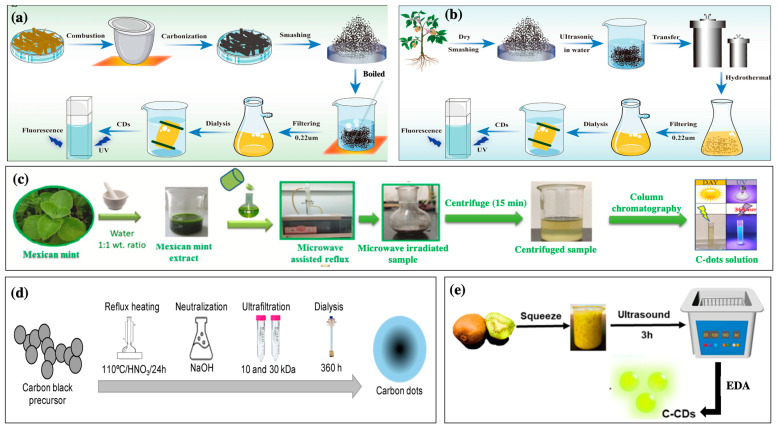
Schematic for synthetic C-dots using various methods: (**a**) high-temperature pyrolysis process of herbal-medicine-derived C-dots [23], (**b**) hydrothermal synthesis process of herbal-medicine-derived C-dots [23], (**c**) synthesis procedure of C-dots derived from Mexican mint (Copyright permission from reference [24]), (**d**) chemical oxidation methodology for synthesis C-dots from carbon black [25], and (**e**) facile synthesis of C-dots from kiwi fruit using ultrasound method [26].

## 3. Cosmetic Properties of Plant Extract-Derived Carbon Dots

### 3.1. Antioxidant Activity

Antioxidants are categorized as enzymatic (peroxidase, ascorbate peroxidase, and catalase) and non-enzymatic (polyphenols, phenolic acids, flavonoids, and ascorbic acid) substances [11]. Enzymatic antioxidants can eliminate free radicals by reducing oxidation products to water in media containing co-factors such as iron, zinc, manganese, and copper [11]. Non-enzymatic antioxidants, such as vitamins, polyphenols, and glutathione, can interrupt ROS chain reactions, thereby preventing oxidation [11]. Plant extract-derived C-dots are classified as non-enzymatic radical scavengers [27].

C-dots can break down reactive nitrogen species (RNS) and hydroxyl radicals (•OH) [28]. First, C-dots possess anti-radical activity towards RNS (i.e., NO• and ${NO}_{2}^{•}$) which can interfere with cellular activity. The antioxidant activity is assessed via evaluating free radical content of a solution containing C-dots and comparing it to that in a solution without C-dots (control). The standard molecule is the 1,1-Diphenyl-2-picrylhydrazyl radical (DPPH•), a nitrogen-centered free radical whose nitrogen has two lone-pair electrons surrounded by three benzene rings. The possible scavenging mechanism involves hydrogen transfer from C-dot surfaces to DPPH• [27] (Figure 2). The presence of carboxyl (–COOH), hydroxyl (–OH), and amino (–NH_2_, –NH–) groups allows the hydrogen transfer and the reduction of DPPH• to DPPH–H [27]. Unpaired electrons on the C-dot surfaces can be delocalized through resonance in aromatic domains or through chemical bond rearrangement. In contrast, the blue fluorescence of C-dots can be quenched by an electron transfer mechanism, in which nitroxide radicals act as electron acceptors [27]. Secondly, C-dots can scavenge the •OH radicals, one of the primary sources of oxidative stress in biological systems, thus preventing oxidative damage to biomolecules (e.g., lipids, proteins, and DNA) caused by •OH radicals [27]. Specifically, C-dots can undergo electron transfer reactions with •OH, resulting in the conversion of the •OH radical into a less reactive molecule that effectively eliminates the radical [27]. In addition, C-dots can participate in redox cycling, where they alternate between the oxidation and reduction states in scavenging reactions. This cycling enhances the efficiency of neutralizing multiple •OH radicals [29,30]. Furthermore, C-dots possess unique photoluminescent properties owing to their quantum confinement and surface states. The excited states of the C-dots can interact with singlet oxygen molecules, which are ROS radicals that are highly responsible for oxidative damage, thus reducing the possibility of singlet oxygen formation [29,30]. The quenching of the excited-state C-dots plays an essential role in their antioxidant properties [28,31].

Recently, several studies have investigated the antioxidant properties of C-dots derived from plant extracts (Table 1). Gudimella et al. (2022) prepared fluorescent C-dots from *Carica papaya* leaves with significant antioxidant and anti-inflammatory properties [32]. The results confirmed that the synthetic C-dots were a significant alternative antioxidant agent with half the maximal effective concentration (EC_50_) of DPPH radical scavenging activity of the C-dots was 27.6 μg/mL [32]. The antioxidant capacity was quantified using a phosphomolybdate assay with an EC_50_ of 23 μg/mL. The oxidation–reduction reactions occurred because of the electron-donating groups on the surface of C-dots, leading to the reduction of molybdenum (VI to V). Li et al. (2021) used biomass *Salvia miltiorrhiza* to produce fluorescent C-dots via the hydrothermal method [33]. The surface of the C-dots containing *S. miltiorrhiza* as a polymer contributed to their high antioxidant capacity. The C-dots scavenged •OH, DPPH• and $O_{2}^{•-}$, with excellent scavenging efficiencies of 71.4, 88.9 and 95.6%, respectively [33].

Skin aging is a natural and gradual process influenced by both intrinsic (hormonal changes, genetics, and metabolism) and extrinsic (environmental exposures and lifestyle choices) factors [11,34,35]. The signs of aging include wrinkles, fine lines, uneven pigmentation, and loss of skin elasticity, which are caused by key factors such as genetics, the degradation of collagen and elastin, the loss of hyaluronic acid, oxidative stress, glycation, and inflammation (Figure 3) [34]. It is well known that antioxidants (e.g., vitamins C, E, A, flavonoids and phenolics, and hydroxycinnamates) can effectively inhibit these oxidation pathways by strongly stimulating old keratinocytes, eliminating ROS radicals, increasing collagen synthesis, and regulating gene expression to improve facial skin conditions [11,34]. Therefore, C-dots with antioxidant properties can effectively scavenge ROS radicals and prevent cell damage, helping maintain the health and function of cells, which is essential for the overall anti-aging effects. Moreover, C-dots are generally considered biocompatible, indicating that they can interact with biological systems without causing significant side effects. Biocompatibility is key factor for promising therapeutic applications, including those related to anti-aging. Recently, fluorescent carbon dots developed from tannic acid using the microwave-assisted pyrolysis method exhibited high biocompatibility (cell viability of 99.7 ± 0.8%) and excellent free ROS radical scavenging (82.8 ± 4.3%) [36]. The tannic acid-derived C-dots (10 μg/mL) significantly inhibited skin aging-related tyrosinase, elastase, and collagenase by 44.2 ± 1.3%, 52.6 ± 1.0%, and 77.6 ± 4.8%, respectively. The results suggested that tannic acid with low toxicity and superior anti-aging and antioxidant properties could be an excellent anti-aging material for cosmetic applications [36]. Moon et al. (2020) produced C-dots from *Opuntia humifusa* using a microwave-assisted method (800 W for 10 min) [10]. Synthetic C-dots exhibited outstanding antioxidant and anti-pollutant activities by suppressing AhR degradation, ROS production, and MMP-9 and COX-2 expressions in human keratinocytes (HaCaT cells), suggesting potential cosmeceutical properties [10]. Plant extract-derived C-dots with potential antioxidant properties are excellent candidates for anti-aging applications.

### 3.2. Anti-Inflammatory Activity

After long-term exposure to external harmful factors (UV irradiation, pollutants, and smoke), ROS radicals are generated on the skin, mediating inflammatory responses that can kill a significant number of skin cells [35]. Generally, ROS radicals downregulate receptor protein tyrosine phosphatases’ (RPTPs) activity, thereby enhancing the expression of phosphorylated receptor tyrosine kinases (RTKs), activating downstream signaling pathways (e.g., transforming growth factor (TGF-β), transcription factor activator protein-1 (AP-1), subsequent nuclear factor-κB (NF-κB), and the activation of mitogen-activated protein kinase (MAPK)) [11,37]. Additionally, the external factors upregulate the expression of tumor necrosis factor-α (TNF-α), which is important in the pro-inflammatory process, resulting in the inhibition of collagen synthesis and stimulation of matrix metalloproteinase (MMP-9) production [11]. It was also found that long-term exposure to UV radiation can upregulate the expression of other cytokines, such as cysteine-rich protein (CCN1) and interleukins (IL-1, IL-6, and IL-8), which promote aging-related and skin-inflammatory processes [11,38].

The potential anti-inflammatory effects of plant extract-derived C-dots can be attributed to their ability to scavenge ROS radicals, modulate immune responses, and interact with inflammatory signaling pathways. In particular, the C-dots can effectively scavenge ROS radicals and downregulate pro-inflammatory mediators (e.g., TNF-α, IL-1α, IL-1β, IL-2, IL-6, IL-8, IL-12, and IFN-γ receptors), which are useful for improving the immune system [10,39,40]. Deng et al. (2023) reported the powerful anti-inflammatory activities of C-dots produced from *Broccoli* water extract using a hydrothermal method (180 °C and 8 h) [41]. The synthetic C-dots remarkably reduced the expression of IL-6 and TNF-α as the concentration of C-dots increased, compared to the control group [41]. C-dots showed a good ability to regulate NO production by significantly reducing ${NO}_{3}^{-}$ in zebrafish after treatment with C-dots for 2 h [41]. In addition, the synthetic C-dots upregulated the expressions of superoxide dismutase (SOD) and glutathione peroxidase (GPX-4), which, in turn, scavenge excessed ROS and reduce inflammation [41]. Therefore, plant extract-derived C-dots can effectively control inflammation-induced skin conditions (e.g., redness, irritation, and premature aging) and even certain skin conditions (such as acne, rosacea, and eczema). Studies on the anti-inflammatory activity of C-dots are presented in Table 1.

Acne is often caused by inflammation due to an immune response triggered by excess oil and bacteria [42]. C-dots with anti-inflammatory properties can help alleviate the inflammation associated with acne lesions, promoting faster healing and reducing the risk of scarring [40,43]. Secondly, controlling inflammation can prevent aging signals, because chronic inflammation can contribute to premature aging by breaking down collagen and elastin fibers [11]. As anti-inflammatory agents, C-dots can potentially slow this process and contribute to maintaining skin elasticity and firmness [36,44]. In addition, inflammation can compromise the natural barrier function of the skin, inducing moisture loss and vulnerability to external irritants [45]. C-dots enhance skin barrier function, helping the skin stay hydrated and protected [40,45].

### 3.3. UV Absorption Properties

One of the critical properties of C-dots is their UV absorption behavior, which exhibits interesting optical properties that are useful for various applications, including sensing, bioimaging, optoelectronics, and cosmetics [9,46]. The UV absorption of C-dots involves electronic transitions within the nanoscale carbon-based structure, which varies depending on their size of C-dots, surface functionalization, and synthesis method [9,47]. The absorption bands of the carbon core are attributed the π–π* transition of aromatic C=C bonds or the n–π* transition of C=O/C=N bonds [47]. The absorption properties of C-dots are affected by the types and content of surface groups, the size of π-conjugated domains, and the oxygen/nitrogen content in carbon cores [47]. The exited-state intramolecular proton transfer through O–H–O and O–H–N tunnels and ample conjugated structures are responsible for the barrier property of C-dots [47]. In addition, due to quantum confinement, smaller-sized C-dots have a larger bandgap and energy separation between electronic energy levels, resulting in higher-energy electronic transitions that lead to UV absorption. Ezati et al. (2022) demonstrated the ultraviolet-blocking property of pectin/carbon dots fluorescent film [47]. The C-dot-added composite films exhibited a light UV absorption at wavelength of 230–320 nm and prevented more than 90% of UV transmission at 280 nm [47]. Min et al. (2023) successfully applied gelatin/poly(vinyl alcohol)-based functional films integrated with spent coffee ground-derived carbon dots and grapefruit seed extract for active packaging application based on the UV-absorption properties of C-dots [48]. The C-dot-added film completely blocked UV rays but sacrificed a little transparency of the film [48]. The results suggested that the intramolecular protein transport through O–H–N, O–H–O tunnels and appropriate junction structures resulted in the UV protection properties of film-containing C-dots [48]. Overall, the UV absorption properties can be harnessed for UV-blocking materials, sensors, and photovoltaic devices. For instance, C-dots with strong UV absorption can be incorporated into sunscreens or other materials to provide UV protection (Table 1). 

**Table 1 nanomaterials-13-02654-t001:** Plant extract-derived C-dots with high antioxidants, anti-inflammatory properties, and UV protection for further cosmetic application.

Plant Sources	Production Techniques	Dimensions (nm)	Cosmetic Properties		Ref.
*Beta vulgaris*	Hydrothermal (160 °C for 8 h)	3–5	Antioxidant	DPPH radical scavenging 94.5%	[49]
*Broccoli* water extract	Hydrothermal (160 °C for 8 h)	5–15	Antioxidants	DPPH radicals scavenging 47.5% ABTS radicals scavenging 52.6% Low cytotoxicity (100% cell viability at 100 μg/mL C-dots)	[41]
			UV protection	UV absorption at 265 nm	
*Carica Papaya* leave	Sand bath method (160 °C for 24 h)	2–20	Antioxidant	DPPH free radical scavenging 85%	[32]
			Anti-inflammatory	Inhibition the lysis of overloaded red blood cells (RBC) 91%	
Red Korean ginseng	Microwave-assisted method (700 W and 14 min)	1–4	Antioxidant	DPPH radical scavenging 85.4% H_2_O_2_ free radical scavenging 87% Low cytotoxicity (80% cell survival at 100 μg/mL C-dots)	[50]
*Salvia miltiorrhiza*	Hydrothermal (160 °C for 6 h)	1.53–16.94	Antioxidant	SOD scavenging 95.6% Hydroxyl radical scavenging 71.4% DPPH radical scavenging 88.9% Low cytotoxicity (90% cell survival at 2 mg/mL C-dots)	[33]
			UV protection	UV absorption peaks at 280 and 325 nm	
*Gynostemma*	Calcination	~2.49	Antioxidant	H_2_O_2_ radical scavenging 80%	[51]
			UV protection	UV absorption peak at 268 nm Long-term stability (30 d) at room temperature	
*Ocimum tenuiflorum*	Hydrothermal (200 °C and 20 h)	1–3	Antioxidant	DPPH radical scavenging 87% Low cytotoxicity (95% cell survival at 200 μg/mL C-dots)	[52]
			UV protection	UV absorption peaks at 254 and 315 nm	
*Centella asiatica*	Microwave-assisted method (800 W and 10 min)	2.97–5.26	Antioxidant	DPPH radical scavenging 78.97% H_2_O_2_ scavenging 89.47%	[53]
			UV protection	UV absorption peak at 213 nm	
			Anti-inflammatory	Inhibition of protein denaturation of 85.84%	
*Annona squamosa* leaves	Hydrothermal (140 °C and 10 h)	2–3	UV protection	UV absorption peaks at 223 and 281 nm	[54]
			Anti-inflammatory	Inhibition they lysis of overloaded RBC 20–50%	
*Rose-heart radish*	Hydrothermal (180 °C and 3 h)	1.2–6	UV protection	UV absorption peak at 282 nm	[55]
*Red cabbage*	Hydrothermal (200 °C and 36 h)	2–7	Antioxidant	DPPH radical scavenging 61% •OH radical scavenging 56%	[56]
			UV protection	UV absorption peak at 365 nm	
*Trapa bispinosa* peel	Pyrolysis	5–10	UV protection	UV absorption peak at 538 nm	[57]
*Solanum lycopersic*	Hydrothermal (160 °C for 3 h)	~9	Antioxidant	DPPH radical scavenging 63.8%	[1]
			UV protection	UV absorption peak at 265 nm	
*Ananas comosus*	Hydrothermal (200 °C for 6 h)	1.0–4.0	Antioxidant	DPPH radical scavenging 50.2% H_2_O_2_ scavenging 93.4%	[58]
*Echinops persicus*	Hydrothermal (200 °C for 10 h)	4–6	Antioxidant	DPPH radical scavenging 95%	[59]
*Opuntia humifusa*	Microwave-assisted method (800 W and 10 min)	5–10	Antioxidant	DPPH radical scavenging 94% •OH radical scavenging 81%	[10]
*Leathesia difformis*	Hydrothermal (180 °C for 4 h)	4–15	UV protection	UV absorption peak at 365 nm Long-term photostability UV blocking 99.96%	[60]
*Dunaliella salina*	Hydrothermal (200 °C for 3 h)	3.0–3.5	UV protection	UV absorption peak at 270 nm SPF values of 35	[61]

## 4. Plant Extract-Derived C-Dots Used in Cosmetics

Recently, C-dots have been used in cosmetic formulations to improve the appearance of the skin due to their superior properties, including luminescent, antioxidant, anti-inflammatory, and UV-absorbing properties [36,39,48,62,63]. C-dots can be applied in a variety of cosmetics, such as fluorescent pigments, highlighters and illuminators, anti-UV, color-changing cosmetics, nail polish, and multi-functional cosmetics. First, the luminescent properties allow the C-dots to emit bright and colorful light when exposed to specific wavelengths; therefore, they can be used as fluorescent pigments to achieve brilliant and colorful effects [63,64,65]. The incorporation of C-dots into cosmetics (e.g., lipstick, eye shadow, and nail polish) is expected to enhance attractive and dynamic color effects. Additionally, the C-dots’ luminescent properties can create a subtle shimmer or bolder highlight, adding a colorful and glamorous dimension to makeup [62]. Therefore, plant extract-derived C-dots should be considered as the main ingredient in illuminating powders to create a bright and radiant effect on the skin. In particular, C-dots can be designed to exhibit color-changing properties based on temperature, pH, and light exposure [66]. Incorporating such color-changing C-dots into lipstick, blush, or nail polish can create products that change color based on factors such as skin pH or environmental conditions, providing a unique and customizable makeup experience [45,67]. Second, plant extract-derived C-dot extracts with anti-UV properties can also be applied as sunscreen ingredients [9,47,48]. Ideal UV absorbing and scattering properties of C-dots effectively protect the skin from harmful UV radiation while also being able to provide a color changing effect under UV light. Third, C-dots derived from plant extracts should be integrated into multifunctional cosmetic products (e.g., eyeshadows and lipsticks) to provide color and moisturizing and anti-aging benefits, as well as other skin care thanks to their versatile properties. Excellent antioxidant and anti-inflammatory properties inherited from plant extracts can help control skin aging and acne; therefore, C-dots should be included in anti-aging and acne treatment products [36,39]. It can be seen that incorporating C-dots derived from plant extracts into colorful cosmetics offers an exciting opportunity to enhance the aesthetics and functionality of makeup products, giving attractive options in the world of beauty and cosmetics. However, it is essential to ensure the safety and regulatory compliance of these products before widespread commercial use.

A study developed new carmine cochineal (Car) C-dots with a powerful moisture retention capability [45]. The results showed that Car C-dots absorbed UV irradiation at wavelengths of 213 and 283 nm, which are attributed to π–π* and n–π*, respectively [45]. The photoluminescence spectra showed an emission at 558 nm under UV light excitation 283 nm. The moisture retention capacity of the Car C-dots under high-humidity conditions was equivalent to that of glycerin, which is commonly used as a hygroscopic agent (approximately 78% and 84%, respectively, after 48 h). Human umbilical vein endothelial (HUVEC) cells were more than 80% viable after exposure to Car C-dots at concentrations reaching 300 μg/mL, indicating the low cytotoxicity of Car C-dots. The results also showed that Car C-dots caused low hemolytic toxicity to red blood cells (RBCs), allowing subsequent blood applications. Additionally, the Car C-dots were successfully applied to moisturizing lipsticks (Figure 4) [45]. The moisture retention capacity of a volunteer’s skin was significantly improved after applying Car C-dot-based moisturizing lipstick. C-dots possess a cross-link-enhanced (CE) effect, which enhances the ability of their surface functional groups to bind water molecules during the cross-linking process [45,68]. Furthermore, the surface functional groups of the C-dots exhibited polymer-like properties and could effectively lock water molecules through water absorption and swelling [45,68]. Owing to their unique moisturizing properties due to the combination of polymer swelling and the CE effect, C-dots have significant potential in cosmetics.

Kim et al. (2022) synthesized UV-protective C-dots from sea cauliflower (*Leathesia difformis*) using a one-pot hydrothermal approach (Figure 5) [60]. A clear yellowish-brown C-dot solution was obtained after the hydrothermal reaction. The solution had two sharp peaks at 290 and 294 nm under UV irradiation, which were attributed to the π–π* transition of the C=C bonds in the aromatic sp^2^ domain [60]. The photostability of the initially synthesized C-dots was determined by measuring their photoluminescence intensities after exposure to 20 mW/cm^2^ UV in a dark room for 360 min. The photoluminescence intensity of the C-dots decreased by approximately 82% during the first 180 min and remained stable for the next 180 min. After 1.4 years of storage in a refrigerator, the C-dots exhibited maximum photoluminescence intensities of approximately 73% compared with the initial values 1.4 years earlier. The UV-blocking ratio of the C-dot solution evaluated from the transmittance values was approximately 99.96 ± 0.08%. The synthetic C-dots exhibited long-term stability and a high UV blocking ratio, indicating their potential for sunscreen applications.

Ghatzimitakos et al. (2020) also investigated the sun protection properties of C-dots synthesized from *Dunaliella salina* (Figure 6) [61]. The fabricated C-dots exhibited a weak shoulder peak at a wavelength of 270 nm under UV irradiation, which was attributed to the n–π* and π–π* transitions of the –C–O bonds in the carboxyl group or the π–π* transitions or aromatic –C=C bonds. The C-dots were stable after 2 h of UVB irradiation (290–320 nm), indicating good optical properties for potential use in sunscreens. A sunscreen formulation containing 11 mg/mL of the synthetic C-dots had a sun protection factor (SPF) value of 37. In addition, the C-dots were non-toxic to the cells at concentrations of 100–600 μg/mL. The ability of C-dots to protect cells from UV irradiating was assessed by pre-treating HaCaT cells with C-dots (100–200 μg/mL) and then irradiation the cells with UV light. The pre-treatment of C-dots increased the cell viability (23–34%), compared to the control group without pre-treatment with C-dots. Therefore, the C-dots can be considered active ingredients for further developing a multifunctional sunscreen formulation.

## 5. Toxicity Assessment of C-Dots Used in Cosmetic Formulation

Although the prospective properties of C-dots using in cosmetic are broad, it is also necessary to consider the potential risks due to the interactions between C-dots and biological systems. Ideally, once C-dots have penetrated into the skin layers, C-dots should perform their required functions and then be excreted from the body without any adverse effects. Therefore, it is important to comprehensively analyze the toxicity of different types of plant extract-derived C-dots from the perspective of toxicology. To assess the safety and in vitro and in vivo cytotoxicity, the biocompatibilities of C-dots are often determined through toxicity studies, including cell proliferation, DNA damage, oxidative stress, apoptosis, and necrosis. Skin irritation and sensitization testing (e.g., the Draize test and local lymph node assay) should be conducted to determine whether C-dots can induce skin irritation and allergic reactions. From the results of these in vitro and in vivo evaluations, the manufacturers should establish quality control measures to ensure the consistent production of C-dots with minimal impurities and variation in toxicity. In addition, a regulatory compliance is required to ensure that the cosmetic product containing C-dots complies with local and international regulations for cosmetics, including safety and labeling requirements.

### 5.1. In Vitro Toxicity

In vitro toxicity is generally evaluated by cell proliferation during in vitro assays (e.g., MTT, WST-1, and CCK-8 assays). Many studies have shown that cell viability negligibly decreases with an increase in the amount of plant extract-derived C-dots added. Son et al. (2021) evaluated the cytotoxicity of antioxidant and anti-aging C-dots derived from tannic acid (100–500 μg/mL) using a CCK-8 assay in human melanoma skin cells (Hs 294 T) [36]. The results showed that even high C-dots concentrations (500 μg/mL) exhibited high cell viability (99.7 ± 0.8%) in Hs 294 T cells, suggesting low cytotoxicity. This study confirmed that C-dots are non-hazardous and can be anti-aging ingredients [36]. C-dots derived from *Broccoli* water extract exhibited very low cytotoxicity for A549 and 293T cells with cell viability of about 100% and over 80% at 100 and 500 μg/mL concentrations after 24 h of exposure, respectively [36]. The C-dots derived from carmine cochineal (Car C-dots) for lipstick applications also exhibited low toxicity even at a high concentration of about 300 μg/mL (more than 80% HUVEC cell survival) [45]. The Car C-dots, with low toxicity and high moisture retention capability, are a potential active ingredient for lipstick application [45]. It can be seen that most plant extract-derived C-dots have good biocompatibility and low toxicity.

### 5.2. In Vivo

In vivo toxicity studies involving plant extract-derived C-dots aim to evaluate their safety and potential adverse effects when introduced into living organisms. These studies include administering C-dots to animal models (e.g., mice, rats, or zebrafish embryos) and evaluating their physiological, biochemical, and behavioral effects in a specified period [69]. The biocompatibility of the *Broccoli*-water-extract-derived C-dots was evaluated in-vivo with zebrafish embryos [36]. Overall survival and hatching rates were higher than 85%, even at the highest C-dot concentration, showing that the C-dots did not cause remarkable mortality or delayed hatching [36]. Wei et al. (2019) also evaluated the biocompatibility of *Gynostemma* extract-derived C-dots using in-vivo cytotoxicity on zebrafish embryos [51]. The results showed that the embryo survival rate and hatching rate were about 90%, and the malformation rate was less than 10%, indicating the excellent biocompatibility of the synthetic C-dots [51].

In vivo toxicity also requires assessing the interaction of C-dots with the extracellular components and the matrix of various cellular components of the tissue structure in a given organ [69]. During systemic transport, C-dots can circulate directly in the body through the blood, enter the spleen and kidney, and finally pass through the kidney filtration system to be eliminated from the body [69]. Tao et al. (2012) conducted in vivo toxicology of C-dots (20 mg/kg) on female Balb/c mice for 90 days [70]. The careful autopsies on days 1, 7, 20, 40, and 90 after the injection of C-dots showed no organ damage and no side effects in the injected group compared to the control group [70]. Huang et al. (2013) reported that C-dots could be rapidly and effectively eliminated from the body through urine after injection into mice [71]. Overall, the toxicity of plant extract-derived C-dots in vitro and in vivo is negligible, and they have good prospects for cosmetic application.

## 6. Behavior of C-Dots in Delivery Systems

Even though plant extract-derived C-dots possess potential cosmetic properties (antioxidant, anti-inflammatory, and UV absorption activities), they still face long-term core challenges in the cosmetic industry, specifically instability, low skin permeability, and risk of toxicity [69,72]. Therefore, incorporating C-dots into a delivery system is expected to overcome these drawbacks that can control various parameters in the cosmetic manufacturing process, such as better stability, delivery efficacy, long lifetime, high biocompatibility, and site specificity. Multiple delivery systems can be used to ensure their stability, effectiveness, and safety when incorporating C-dots into cosmetic products. The typical delivery systems containing carbon dots for cosmetic applications include nanoemulsions, liposomes, solid lipid nanoparticles (SLNs), and hydrogels (Figure 7 and Table 2) [73,74,75]. First, C-dots can be dispersed in oil-in-water (O/W) or water-in-oil (W/O) nanoemulsions, which are highly stable and have a small droplet size due to their small size, which leads to valuable properties such as high surface area per unit volume, robust stability, optically transparent appearance, and tunable rheology [74,76]. Second, C-dots can be encapsulated into lipid-based vesicles, liposomes, which are spherical vesicular structures that are physiologically present in the spinous layer of the epidermis [73,74]. Third, C-dots can be loaded into SLNs with a similar structure to nanoemulsions, but the oil phase is wholly or partially solidified. The SLN system promotes skin hydration via stratum corneum occlusion, which prevents water loss by evaporation, and via the reinforcement of the skin’s lipid–film barrier, which occurs through the adhesion of the NPs to the stratum corneum [77,78]. Fourth, C-dot-based hydrogel systems are prepared from biopolymer molecules with three-dimensional network structures that contain a relatively large amount of water [75]. The core of a hydrogel is a polymeric channel system, which may be formed through the physical or chemical cross-linking of homopolymers or copolymers, resulting in swelling when subjected to an aqueous surrounding [75,79]. Hydrogels have been used for a long time in dermatology and cosmetic fields due to their biocompatibility, their good spreading and adherence to the site of application, and their long resident time on the site of action facilitating diffusion in the skin [75,79].

These delivery systems are considered active vectors in C-dot carriers regarding their ability to improve their solubility and stability, thus effectively delivering incorporated C-dots to specific targets and providing sustained C-dot release [73,74]. In addition, these delivery systems are composed of physiological lipid membranes that can reduce the danger of acute and chronic toxicity [73,76,77]. Furthermore, these delivery systems can remain stable against various chemical and environmental changes, thereby increasing the stability of the C-dots in cosmetic formulations, consequently enhancing the performance of cosmetic products [74,75]. Using delivery systems as effective vehicles protects reactive or sensitive compounds against environmental conditions, including light, temperature, pH, humidity, and oxygen. In addition, these delivery systems with biphasic characteristics can improve the solubility of poorly water-soluble C-dots [80]. The lipid bilayers are likely to enhance the water solubility of C-dots and be homogenously dispersed in cosmetic formulations for the significantly increased bioavailability of C-dots. Moreover, these delivery systems, with their small sizes and spherical bilayer structures, can mimic the cell membrane constituents and improve dermal and transdermal active-C-dot delivery [74,75].

**Figure 7 nanomaterials-13-02654-f007:**
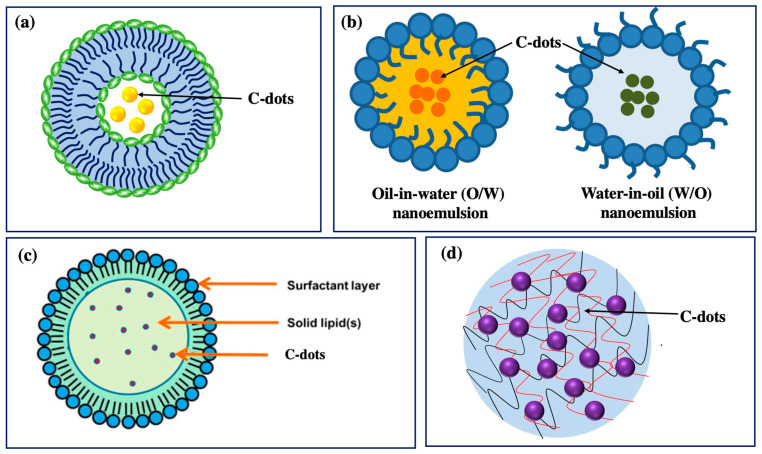
Delivery systems based on C-dots for cosmetic application: (**a**) liposome-encapsulated C-dots; (**b**) nanoemulsion-loaded C-dots (copyright permission from reference [81]); (**c**) solid lipid nanoparticle-loaded C-dots (copyright permission from reference [82]); and (**d**) hydrogel-loaded C-dots [83].

**Table 2 nanomaterials-13-02654-t002:** Typical delivery systems based on C-dots for future applications [73,74,75,76,78].

Delivery System	Positive Aspects	Negative Aspects	Current Studies
Liposomes	Increased efficacy and stability Reduced toxicity Biocompatible and biodegradable Enhanced skin permeability to the dermal layer Site avoidance effect	Low solubility High production cost Inadequate stability Leakage of drug Osmotically sensitive Occasionally, oxidation and hydrolysis reaction	Demir et al. (2021) developed C-dots and curcumin-loaded CD44-targeted liposomes for imaging and tracking cancer chemotherapy [84]. Ren et al. (2019) encapsulated near-infrared fluorescent C-dots into liposomes that help improve drug release, enhance anticancer efficacy, and reduce side effects of C-dots [85]. Guan et al. (2018) encapsulated C-dots into liposomes to increase the stability and biocompatibility of C-dots for targeting recognition to HepG2 cells [86].
Nanoemulsions	Improved rate of absorption Suitable for both lipophilic and hydrophilic delivery systems Nontoxic and nonirritant Do not cause side effects such as coalescence, sedimentation, flocculation, and creaming	Its stability is disturbed by environmental parameters (pH and temperature) A high concentration of surfactants/cosurfactants is necessary for the stabilization of the nanodroplets Limited solubility for high-melting substances	Yi et al. (2022) successfully stabilized surface-tailor C-dots in an O/W emulsion. The encapsulated C-dots showed high fluorescence emission and effective adsorption of heavy metal ions and organic dyes [87]. Ettefaghi et al. (2021) developed bio-nanoemulsion fuel based on graphene quantum dots to optimize fuel consumption, enhance the performance of diesel engines, and reduce pollutant emissions [88].
SLNs	Improved stability and biocompatibility Controlled release and increased bioavailability of C-dots Increased skin hydration and skin permeability of C-dots Quickly scale up production	Poor encapsulation efficiency High water content Burst release can take place Unpredictable gelation tendency	Kim et al. (2023) prepared a drug delivery system by encapsulating pH-sensitive C-dots and fatty acids into SLNs with minimal drug loss. The obtained SLN systems showed minimal encapsulated drugs in the blood (<10–20%) and enhanced tumor growth inhibition efficacy [89]. Arduino et al. (2020) incorporated polyethylene glycol (PEG)-Pt(IV) prodrugs and luminescent C-dots in SLNs. The SLNs showed good stability in aqueous media and high permeability into brain tumor animal cells, suggesting the potential antitumor efficiency of SLNs [90].
Hydrogels	Biocompatible with A large number of water-making delivery systems and suitable for applications involving biological systems Slowly and steadily release C-dots over timeGood biodegradable systems help to reduce environmental impacts Reduced side effects	Sensitive to changes in temperature and pH High swelling and degradation affect the stability and functionality of hydrogels Low encapsulation efficacy High production cost	Panda et al. (2022) developed a hydrogel system based on papaya-derived C-dots. The delivery system showed significant antibacterial activity, high drug release efficiency at B16F10 melanoma cells, and fewer side effects [91]. Turk et al. (2021) developed novel injectable pH-sensitive C-dots and hydroxyapatite hydrogel delivery system by self-crosslinking approach. The hydrogel system showed low toxicity and a good ability to transport drugs to specific cancer cells targeted at a specifically acidic pH [92]. Wang et al. (2017) reported the photothermal-chemo therapy of chitosan C-dots loaded in a hydrogel delivery system. The hydrogel system showed high colloidal stability and loading efficiency, and stable fluorescence. The system could permeate into the tumor and release drug molecules efficiently to inhibit tumor growth [93].

## 7. Conclusions

The use of plant extract-derived C-dots in cosmetics has opened possibilities for bridging sustainability, nanotechnology, and natural ingredients. This review meticulously discusses the synthesis methods and potential aesthetic properties of plant extract-derived C-dots, emphasizing their role in shaping the future of cosmetics. These C-dots possess the inherent attributes of plants, such as their antioxidant, anti-inflammatory, and biodegradable properties, and provide a distinct advantage over conventional synthetic materials. Their tunable optical properties, achieved through quantum confinement effects, are promising for applications such as pigmentation modulation and sunscreen protection, thereby expanding the possibilities of cosmetic formulations. By enhancing their properties and sustainability, C-dots can potentially revolutionize the cosmetics industry. However, challenges and questions remain regarding emerging technologies. Further research is necessary to comprehensively assess the stability and long-term effects of incorporating these C-dots into cosmetic products. These C-dots should be incorporated into delivery systems (liposomes, nanoemulsions, SLNs, and hydrogels) to overcome recent challenges, such as low stability, low skin permeability, and risk of toxicity. Standardizing the synthesis methods, characterization techniques, and regulatory considerations is vital for ensuring consistent quality and consumer confidence.

## Figures and Tables

**Figure 2 nanomaterials-13-02654-f002:**
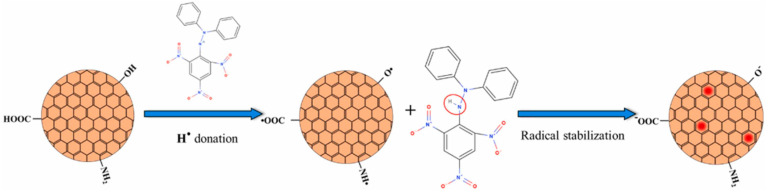
Mechanism of DPPH• reduction by antioxidant C-dots in aqueous media. Copyright permission from reference [27].

**Figure 3 nanomaterials-13-02654-f003:**
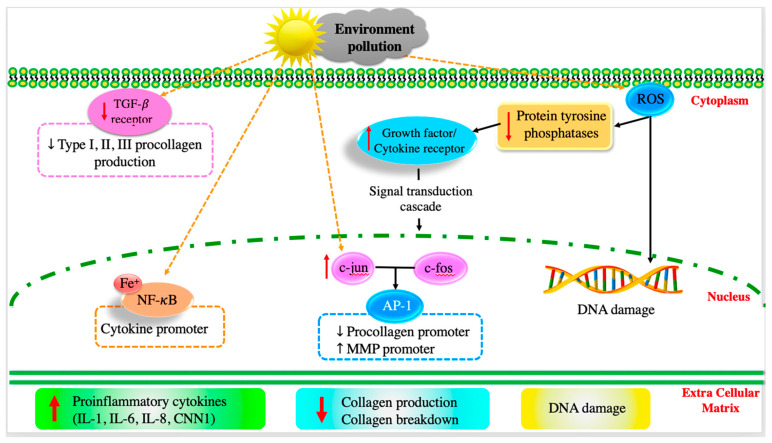
Intracellular mechanism of skin ageing. Copyright permission from reference [11].

**Figure 4 nanomaterials-13-02654-f004:**
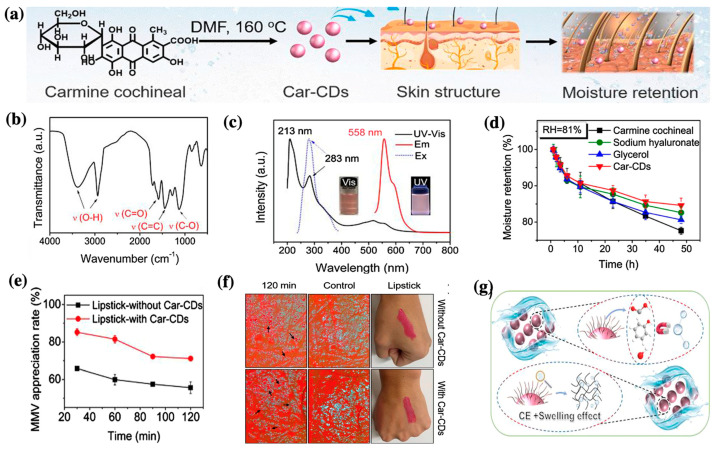
(**a**) Schematic of the preparation and structure of Car C-dots; (**b**) Fourier transform infrared (FT-IR) spectra of Car C-dots; (**c**) UV-Vis absorption and photoluminescent spectra of Car C-dots; (**d**) the relationship between moisture retention and time at RH = 81% for different samples; (**e**) changes of hand skin moisture content after application of the moisturizing lipstick; (**f**) changes in moisture content in the hand skin after applying the moisturizing lipstick; and (**g**) the possible mechanism of the moisture retention ability of Car C-dots. Modified from reference [45].

**Figure 5 nanomaterials-13-02654-f005:**
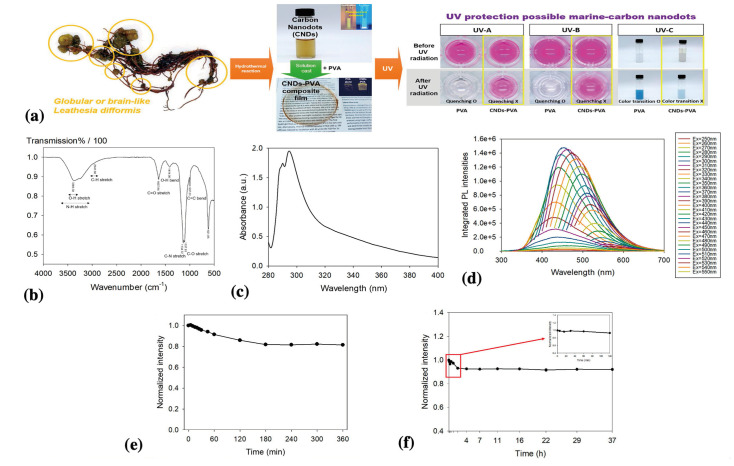
(**a**) Schematic of the synthesis of C-dots from sea cauliflower and their UV protection capabilities; (**b**) FT-IR spectra of C-dots; (**c**) UV-Vis absorption spectra of C-dots; (**d**) photoluminescent emission spectra in the range of wavelength from 250 nm to 550 nm; (**e**) photostability of C-dots solution under UV irradiation for 360 nm; and (**f**) photostability of C-dots solution under UV irradiation for 37 h 1.4 years after the initial synthesis. Copyright permission from reference [60].

**Figure 6 nanomaterials-13-02654-f006:**
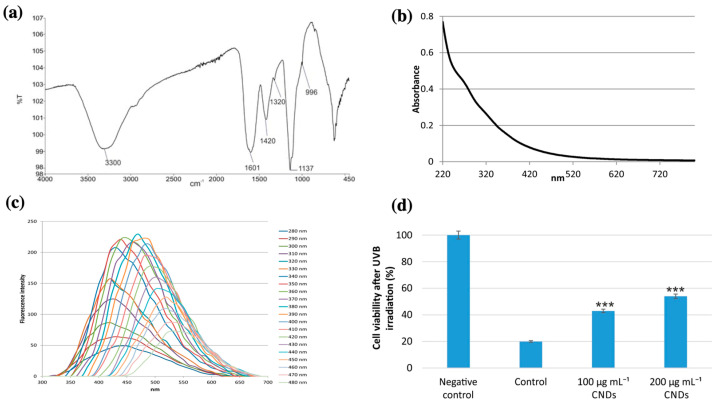
(**a**) FT-IR spectra of *Dunaliella salina* extract-derived C-dots; (**b**) UV-Vis spectra of the synthetic C-dots; (**c**) photoluminescent emission spectra of the C-dots at excitation wavelengths between 280 nm and 480 nm; and (**d**) cell viability (%) of HaCaT cells in negative control (without UV irradiation), control (cells irradiated with UV rays and without C-dots), and cells pre-treated with 100 μg/mL and 200 μg/mL C-dots (*** denotes statistically significant differences at *p* < 0.001). Modified from reference [61].

## Data Availability

Not applicable.

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
