# Peer review of "Plant Extract-Derived Carbon Dots as Cosmetic Ingredients"

_nanomaterials, 2023, doi:10.3390/nano13192654_

Round 1
Reviewer 1 Report
Dear authors
This is an interesting work on a field that is underdeveloped in the cosmetic industry.
The review is thorough and well written though extensive English language correction needs to be done.
The most important aspect that this review lacks is an understandable part concerning the safety and possible toxicity of Carbon Dot systems. As safety is the most important part of cosmetic formulations, there should be a separate chapter analyzing safety/toxicity data from the literature and critically discussing upon this.
Extensive English language correction needs to be done - both in grammar and syntax mistakes.
Eg.
By leveraging the inherent attributes of plants, such as their antioxidants...
...skin aging, treating skin conditions caused by inflammation, and sunscreen...
Acne is commonly attributed to inflammation due to an immune response triggered by excess oil and bacterial..
Author Response
Response sheet
Manuscript ID: nanomaterials-2611397
Type of manuscript: Review
Title: Plant-derived Carbon Dots as Cosmetic Ingredients
Authors: Le Thi Nhu Ngoc, Ju-Young Moon *, Young-Chul Lee *
Reviewer 1
Dear authors
This is an interesting work on a field that is underdeveloped in the cosmetic industry.
The review is thorough and well written though extensive English language correction needs to be done.
- The most important aspect that this review lacks is an understandable part concerning the safety and possible toxicity of Carbon Dot systems. As safety is the most important part of cosmetic formulations, there should be a separate chapter analyzing safety/toxicity data from the literature and critically discussing upon this.
Response: Thank you for your comments. The safety and toxicity data of carbon dots (C-dots) derived from plant extracts play an important role in further applications in cosmetic or biomedical fields. Therefore, the manuscript has fully described this aspect in the revision. Due to the application of C-dots into cosmetics is novelty, the safety assessments of the application are undergoing, and the data are missing. Therefore, this manuscript can only describe the toxicity assessment of C-dots used in cosmetics.
“5. Toxicity assessment of C-dots using in cosmetic formulation
Although the perspective properties of C-dots using in cosmetic are broad, it is also necessary to consider the potential risks due to the interactions between C-dots and biological systems. Ideally, once C-dots are penetrated in the skin layers, C-dots should perform their required functions and then be excreted from the body without any adverse effects. Therefore, it is important to comprehensively analyze the toxicity of different types of plant extract-derived C-dots from the perspective of toxicology. To assess the safety and in-vitro and in-vivo cytotoxicity, the biocompatibilities of C-dots are often determined through toxicity studies, including cell proliferation, DNA damage, oxidative stress, apoptosis, and necrosis. Skin irritation and sensitization testing (e.g., the Draize test and local lymph node assay) should be conducted to determine if C-dots can induce skin irritation and allergic reactions. From the results of these in-vitro and in-vivo evaluations, the manufactures should establish quality control measures to ensure consistent production of C-dots with minimal impurities or variation in toxicity. In addition, a regulatory compliance is required to ensure that the cosmetic product containing C-dots complies with local and international regulations for cosmetics, including safety and labeling requirements.
5.1. In-vitro toxicity
In-vitro toxicity is generally evaluated by cell proliferation during in-vitro assays (e.g., MTT, WST-1, and CCK-8 assays). Many studies have shown that cell viability negligibly decreases as adding plant extract-derived C-dots increases. Son et al. (2021) evaluated the cytotoxicity of antioxidant and anti-aging C-dots derived from tannic acid (100–500 mg/mL) using a CCK-8 assay in human melanoma skin cells (Hs 294 T) [37]. The results showed that even high C-dots concentrations (500 mg/mL) exhibited high cell viability (99.7±0.8 %) in Hs 294 T cells, suggesting low cytotoxicity. This study confirmed that C-dots are non-hazardous and can be anti-aging ingredients [37]. C-dots derived from Broccoli water extract exhibited very low cytotoxicity for A549 and 293T cells with cell viability of about 100 % and over 80 % at 100 and 500 mg/mL concentrations after 24 h of exposure, respectively [37]. The C-dots derived from carmine cochineal (Car C-dots) for lipstick applications also exhibited low toxicity even at a high concentration of about 300 mg/mL (more than 80 % HUVEC cell survival) [46]. The Car C-dots, with low toxicity and high moisture retention capability, are a potential active ingredient for lipstick application [46]. It can be seen that most plant extract-derived C-dots have good biocompatibility and low toxicity.
5.2. In-vivo
In-vivo toxicity studies involving plant extract-derived C-dots aim to evaluate their safety and potential adverse effects when introduced into living organisms. These studies include administering C-dots to animal models (e.g., mice, rats, or zebrafish embryos) and evaluating their physiological, biochemical, and behavioral effects in a specified period [71]. The biocompatibility of the Broccoli water extract-derived C-dots was evaluated in-vivo with zebrafish embryos [37]. Overall survival and hatching rates were higher than 85 %, even at the highest C-dot concentration, showing that the C-dots did not cause remarkable mortality or delayed hatching [37]. Wei et al. (2019) also evaluated the biocompatibility of Gynostemma extract-derived C-dots using in-vivo cytotoxicity on zebrafish embryos [53]. The results showed that the embryo survival rate and hatching rate were about 90 %, and the malformation rate was less than 10 %, indicating the excellent biocompatibility of the synthetic C-dots [53].
In-vivo toxicity also requires assessing the interaction of C-dots with the extracellular and the matrix of various cellular components of the tissue structure in a given organ [71]. During systemic transport, C-dots can circulate directly in the body through the blood, enter the spleen and kidney, and finally pass through the kidney filtration system to be eliminated from the body [71]. Tao et al. (2012) conducted in-vivo toxicology of C-dots (20 mg/kg) on female Balb/c mice for 90 days [72]. The careful autopsies on the 1, 7, 20, 40, and 90 days after injection of C-dots showed no organ damage and no side effects in the injected group, compared to the control group [72]. Huang et al. (2013) reported that C-dots could be rapidly and effectively eliminated from the body through urine after injection into mice [73]. Overall, the toxicity of plant extract-derived C-dots in-vitro and in-vivo is negligible, and they have good prospects for cosmetic application.”
- Comments on the Quality of English Language
Extensive English language correction needs to be done - both in grammar and syntax mistakes.
Eg. By leveraging the inherent attributes of plants, such as their antioxidants...
...skin aging, treating skin conditions caused by inflammation, and sunscreen...
Acne is commonly attributed to inflammation due to an immune response triggered by excess oil and bacterial.
Response: Thank you for your comment. The manuscript was corrected English language by a native speaker. All the grammar and syntax mistakes are carefully corrected according to your suggestion.

Reviewer 2 Report
This review manuscript explores the potential applications of C-dots derived from plant extracts in cosmetics. This paper discusses the synthesis methodologies for generation of C-dots from plant precursors, including pyrolysis carbonization, chemical oxidation, hydrothermal, microwave-assisted, and ultrasonic methods. By leveraging the inherent attributes of plants, such as their antioxidants, anti-inflammatory, and UV protective properties, plant-derived C-dots offer distinct advantages over conventional synthetic materials. These outstanding properties are critical for novel cosmetic applications such as for con-trolling skin aging, treating skin conditions caused by inflammation, and sunscreen. The manuscript is well written with nice English and meaningful insight. Thus, I recommend this work to be accepted for publication.
Author Response
Response sheet
Manuscript ID: nanomaterials-2611397
Type of manuscript: Review
Title: Plant-derived Carbon Dots as Cosmetic Ingredients
Authors: Le Thi Nhu Ngoc, Ju-Young Moon *, Young-Chul Lee *
Reviewer 2:
This review manuscript explores the potential applications of C-dots derived from plant extracts in cosmetics. This paper discusses the synthesis methodologies for generation of C-dots from plant precursors, including pyrolysis carbonization, chemical oxidation, hydrothermal, microwave-assisted, and ultrasonic methods. By leveraging the inherent attributes of plants, such as their antioxidants, anti-inflammatory, and UV protective properties, plant-derived C-dots offer distinct advantages over conventional synthetic materials. These outstanding properties are critical for novel cosmetic applications such as for controlling skin aging, treating skin conditions caused by inflammation, and sunscreen. The manuscript is well written with nice English and meaningful insight. Thus, I recommend this work to be accepted for publication.
Response: Thank you so much for your kind comments. In this review, we describe a novelty topic related to the application of plant extract-derived C-dots for such an interest field, as cosmetics. Given the growing interest in plant-derived ingredients and sustainable cosmetics, we believe that our manuscript will contribute significantly to the knowledge base of the cosmetics industry.

Reviewer 3 Report
In this review, Lee's group summarized the application of CDs in cosmetics. This is an important direction with application value and may serve as an important outlet for the industrialization of CDs in the future, so the selection of topics for the review is very meaningful. The authors have provided a relatively comprehensive overview of the relevant systems, which can be published in nanomaterials after minor revisions, as suggested below:
1. The authors are requested to distinguish between plant-derived CDs and plant extract-derived CDs, which are quite different in terms of efficacy, cost, and difficulty of purification;
2. Whether the cases in Section 2.1 are appropriate;
3. The authors seem to display the existing CDs that can be applied to cosmetics, but there is no more in-depth analysis and discussion. It is suggested that the authors add the structure of CDs related to the properties, and provide guiding insights around the conformational relationships for the further development of CDs that can be applied to cosmetics;
4. It is recommended to add a chapter on "Behavior of CDs in colloidal systems", including the ability to load other components, stability, etc. To facilitate the authors' reference search, authors can refer to the review by Sun et al. (Small, 2023, 2207048);
5. It is recommended that biocompatibility or toxicity analysis be discussed as a separate section, which is an important advantage of CDs to replace existing efficacy components;
6. The author's overview of cosmetics is not comprehensive, and it is recommended that the author add whether the application of CDs in colorful cosmetics, special effects cosmetics in stage, etc., in addition to skin care products, has been reported;
7. There are important references to be added on the use of carbon nanomaterials in skin care and carbon dots: Langmuir, 2019, 35, 6939-6949;ACS Appl. Bio Mater., 2020, 3, 358-368;Nanoscale, 2023, 15, 275–284;J. Mater. Chem. B, 2018, 6, 7021-7032; Separation and purification technology, 2019, 222, 60-67.
Author Response
Response sheet
Manuscript ID: nanomaterials-2611397
Type of manuscript: Review
Title: Plant-derived Carbon Dots as Cosmetic Ingredients
Authors: Le Thi Nhu Ngoc, Ju-Young Moon *, Young-Chul Lee *
Reviewer 3
In this review, Lee's group summarized the application of CDs in cosmetics. This is an important direction with application value and may serve as an important outlet for the industrialization of CDs in the future, so the selection of topics for the review is very meaningful. The authors have provided a relatively comprehensive overview of the relevant systems, which can be published in nanomaterials after minor revisions, as suggested below:
- The authors are requested to distinguish between plant-derived CDs and plant extract-derived CDs, which are quite different in terms of efficacy, cost, and difficulty of purification;
Response: Thank you for your comment. This review aims to describe the potential of carbon dots synthesized from plant extracts for cosmetic applications. Therefore, the phrase “plant-derived C-dots” is replaced by the upgraded phrase “plant extract-derived C-dots” throughout the text.
- Whether the cases in Section 2.1 are appropriate
Response: Thank you for your comment. The contents in section 2.1 are compiled based on appropriate references cited throughout this section. Therefore, it is believed that the cases in section 2.1. are suitable.
- The authors seem to display the existing CDs that can be applied to cosmetics, but there is no more in-depth analysis and discussion. It is suggested that the authors add the structure of CDs related to the properties, and provide guiding insights around the conformational relationships for the further development of CDs that can be applied to cosmetics.
Response: Thank you for your suggestions. In section 4 “Plant extract-derived C-dots used in cosmetics”, this manuscript includes recently published works demonstrating the utilization of C-dots as cosmetic ingredients. Unfortunately, there are few published studies on this topic, and related studies are ongoing; thus, the section can only be related to three studies. On the other hand, the analyzes required to evaluate the physiochemical properties of a potential cosmetic ingredient have been described in text and figures in each paragraph related to the initial studies. However, the description (text and figure) for each reference work has been upgraded to fully express in-depth outstanding properties of C-dots which are useful for cosmetic applications. Moreover, the manuscript has mentioned all essential properties of C-dots, including antioxidant, anti-inflammatory, and UV protection activities, which are key aspects for selecting appropriate C-dot candidates for further development cosmetics. Therefore, the content presented in section 3 “Cosmetic properties of plant extract-derived carbon dots” is important literature for guiding insights around the conformational relationships for the further development of CDs for cosmetic application.
“A study developed a new carmine cochineal (Car) C-dots with powerful moisture retention capability [46]. The results showed that Car C-dots absorbed UV irradiation at wavelengths of 213 and 283 nm, which are attributed to p-p⁎ and n-p⁎, respectively [46]. The photoluminescence spectra showed an emission at 558 nm under UV light excitation 283 nm. The moisture retention capacity of the Car C-dots under high-humidity conditions were equivalent to that of glycerin, which is commonly used as a hygroscopic agent (approximately 78% and 84% within 48 h, respectively). Human umbilical vein endothelial (HUVEC) cells were more than 80% viable after exposure to Car C-dots at concentrations reaching 300 mg/mL, indicating the low cytotoxicity of Car C-dots. The results also showed that Car C-dots caused low hemolytic toxicity to red blood cells (RBCs), allowing subsequent blood applications. Additionally, the Car C-dots was successfully applied to moisturizing lipsticks (Figure 4) [46]. The moisture retention capacity of a volunteer's skin was significantly improved after applying Car C-dots-based moisturizing lipstick. C-dots possessed a cross-link-enhanced (CE) effect, which enhances the ability of their surface functional groups to bind water molecules during the cross-linking process [46,70]. Furthermore, the surface functional groups of the C-dots exhibited polymer-like properties and could effectively lock water molecules through water absorption and swelling [46,70]. Owing to the unique moisturizing properties due to the combination of polymer swelling and the CE effect, C-dots have high potential in cosmetics.
Figure 4. (a) Schematic of the preparation and structure of Car C-dots, (b) Fourier transform infrared (FT-IR) spectra of Car C-dots, (c) UV-Vis absorption and photoluminescent spectra of Car C-dots, (d) the relationship between moisture retention and time at RH = 81 % for different samples, (e) changes of hand skin moisture content after application of the moisturizing lipstick, (f) changes in moisture content in the hand skin after applying the moisturizing lipstick, and (g) the possible mechanism of the moisture retention ability of Car C-dots. Modified from reference [46].
Kim et al. (2022) synthesized UV-protective C-dots from sea cauliflower (Leathesia difformis) using a one-pot hydrothermal approach (Figure 5) [62]. A clear yellowish-brown C-dots solution was obtained after the hydrothermal reaction. The solution had two sharp peaks at 290 and 294 nm under UV irradiation, which were attributed to the p-p* transition of the C=C bonds in the aromatic sp2 domain [62]. The photostability of the initially synthesized C-dots was determined by measuring their photoluminescence intensities after exposure to 20 mW/cm2 UV in a dark room for 360 min. The photoluminescence intensity of the C-dots decreased by approximately 82% during the first 180 min and remained stable for the next 180 min. After 1.4 years of storage in a refrigerator, the C-dots exhibited maximum photoluminescence intensities of approximately 73% compared with the initial values 1.4 years earlier. The UV-blocking ratio of the C-dots solution was evaluated from the transmittance values was approximately 99.96±0.08%. The synthetic C-dots exhibited long-term stability and a high UV blocking ratio, indicating their potential for sunscreen applications.
Figure 5. (a) Schematic of the synthesis of C-dots from sea cauliflower and their UV protection capabilities, (b) FT-IR spectra of C-dots, (c) UV-Vis absorption spectra of C-dots, (d) photoluminescent emission spectra in the range of wavelength from 250 nm to 550 nm, (e) photostability of C-dots solution under UV irradiation for 360 nm, and (f) photostability of C-dots solution under UV irradiation for 37 h after 1.4 years from the initial synthesis. Copyright permission from reference [62]
Ghatzimitakos et al. (2020) also investigated the sun protection properties of C-dots synthesized from Dunaliella salina (Figure 6) [63]. The fabricated C-dots exhibited a weak shoulder peak at a wavelength of 270 nm under UV irradiation, which was attributed to the n-p* and p-p* transitions of the –C–O bonds in the carboxyl group or the p-p* transitions or aromatic –C=C bonds. The C-dots were stable after 2 h of UVB irradiation (290–320 nm), indicating good optical properties for potential use in sunscreens. A sunscreen formulation containing 11 mg/mL of the synthetic C-dots had a sun protection factor (SPF) value 37. In addition, the C-dots were non-toxic to the cells at concentrations of 100–600 mg/mL. The ability of C-dots to protect cells from UV irradiating was assessed by pre-treating HaCaT cells with C-dots (100-200 mg/mL) and then irradiation the cells with UV light. The pre-treatment of C-dots increased the cell viability (23-34 %), compared to the control group without pre-treatment with C-dots. Therefore, the C-dots can be considered active ingredients for further developing multifunctional sunscreen formulation.
Figure 6. (a) FT-IR spectra of Dunaliella salina extract-derived C-dots, (b) UV-Vis spectra of the synthetic C-dots, (c) photoluminescent emission spectra of the C-dots at excitation wavelengths between 280 nm and 480 nm, and (d) cell viability (%) of HaCaT cells in negative control (without UV irradiation), control (cells irradiated with UV rays and without C-dots), and cells pre-treated with 100 mg/mL and 200 mg/mL C-dots. Modified from reference [63].”
- It is recommended to add a chapter on “Behavior of CDs in colloidal systems”, including the ability to load other components, stability, etc. To facilitate the authors’ reference search, authors can refer to the review by Sun et al. (Small, 2023, 2207048);
Response: Thank you for your comments. A section, “Behavior of C-dots in delivery systems,” has been included in the revised manuscript. This section reviews advanced delivery systems containing plant extract-derived C-dots to overcome the recent drawbacks of C-dots (low stability, low skin permeability, and risk of toxicity). Typical delivery systems include liposomes, nanoemulsions, solid lipid nanoparticles, and hydrogels. This section also describes the advantages and disadvantages of these delivery systems when applied to cosmetic products.
“6. Behavior of C-dots in delivery systems
Even though plant extract-derived C-dots possess potential cosmetic properties (antioxidant, anti-inflammatory, and UV absorption activities), they still face long-term core challenges in the cosmetic industry, specifically instability, low skin permeability, and risk of toxicity [71,74]. Therefore, incorporating C-dots into a delivery system is expected to overcome these drawbacks that can control various parameters in the cosmetic manufacturing process, such as better stability, delivery efficacy, long lifetime, high biocompatibility, and site specificity. Multiple delivery systems can be used to ensure their stability, effectiveness, and safety when incorporating C-dots into cosmetic products. The typical delivery systems containing carbon dots for cosmetic applications include nanoemulsions, liposomes, solid lipid nanoparticles (SLNs), and hydrogels (Figure 7 and Table 2) [75–77]. First, C-dots can be dispersed in oil-in-water (O/W) or water-in-oil (W/O) nanoemulsions, which are highly stable, have small droplet size due to their small size leads to valuable properties such as high surface area per unit volume, robust stability, optically transparent appearance, and tunable rheology [76,78]. Second, C-dots can be encapsulated into lipid-based vesicles, liposomes, which are spherical vesicular structures that are physiologically present in the spinous layer of the epidermal [75,76]. Third, C-dots can be loaded into SLNs with a similar structure to nanoemulsions, but the oil phase is wholly or partially solidified. The SLN system promotes skin hydration via stratum corneum occlusion, which prevents water loss by evaporation, and via the reinforcement of the skin’s lipid-film barrier, which occurs through the adhesion of the NPs to the stratum corneum [79,80]. Fourth, C-dots-based hydrogel systems are prepared from biopolymer molecules with three-dimensional network structures that contain a relatively large amount of water [77]. The core of a hydrogel is a polymeric channel system, which may be formed through physical or chemical cross-linking of homopolymers or copolymers, resulting in swelling when subjected to an aqueous surrounding [77,81]. Hydrogels have been used for a long time in dermatology and cosmetic fields due to their biocompatibility, their good spreading and adherence to the site of application, their long resident time on the site of action facilitating diffusion in the skin [77,81].
These delivery systems are considered active vectors in C-dot carriers regarding their ability to improve their solubility and stability, thus effectively delivering incorporated C-dots to specific targets and providing sustained C-dot release [75,76]. In addition, these delivery systems are composed of physiological lipid membranes that can reduce the danger of acute and chronic toxicity [75,78,79]. Furthermore, these delivery systems can remain stable against various chemical and environmental changes, thereby increasing the stability of the C-dots in cosmetic formulations, consequently enhancing the performance of cosmetic products [76,77]. Using delivery systems as effective vehicles protects reactive or sensitive compounds against environmental conditions, including light, temperature, pH, humidity, and oxygen. In addition, these delivery systems with biphasic characteristics can improve the solubility of poorly water-soluble C-dots [82]. The lipid bilayers are likely to enhance the water-solubility of C-dots and be homogenously dispersed in cosmetic formulations for significantly increased bioavailability of C-dots. Moreover, these delivery systems, with small sizes and spherical bilayer structures, can mimic the cell membrane constituents and improve dermal and transdermal active-C-dots delivery [76,77].
Figure 7. Delivery systems based on C-dots for cosmetic application (a) liposomes encapsulated C-dots, (b) Nanoemulsions loaded C-dots. Copyright permission from reference [83], (c) solid lipid nanoparticles loaded C-dots. Copyright permission from reference [84], and (d) hydrogel loaded C-dots [85].
Table 2. Typical delivery systems based on C-dots for future applications [75–78,80]
|
Delivery system |
Positive aspects |
Negative aspects |
Current studies |
|
Liposomes |
Increased efficacy and stability Reduced toxicity Biocompatible and biodegradable Enhanced skin permeability to the dermal layer Site avoidance effect |
Low solubility High production cost Inadequate stability Leakage of drug Osmotically sensitive Occasionally, oxidation and hydrolysis reaction |
Demir et al. (2021) developed C-dots and curcumin-loaded CD44-targeted liposomes for imaging and tracking cancer chemotherapy [86]. Ren et al. (2019) encapsulated near-infrared fluorescent C-dots into liposomes that help improve drug release, enhance anticancer efficacy, and reduce side effects of C-dots [87]. Guan et al. (2018) encapsulated C-dots into liposomes to increase the stability and biocompatibility of C-dots for targeting recognition to HepG2 cells [88]. |
|
Nanoemulsions |
Improved rate of absorption Suitable for both lipophilic and hydrophilic delivery systems Nontoxic and nonirritant Do not cause side effects such as coalescence, sedimentation, flocculation, and creaming |
Its stability is disturbed by environmental parameters (pH and temperature) A high concentration of surfactants/cosurfactants is necessary for the stabilization of the nanodroplets Limited solubility for high-melting substances |
Yi et al. (2022) successfully stabilized surface-tailor C-dots in an O/W emulsion. The encapsulated C-dots showed high fluorescence emission and effective adsorption of heavy metal ions and organic dyes [89]. Ettefaghi et al. (2021) developed bio-nanoemulsion fuel based on graphene quantum dots to optimize fuel consumption, enhance the performance of diesel engines, and reduce pollutant emissions [90]. |
|
SLNs |
Improved stability and biocompatibility Controlled release and increased bioavailability of C-dots Increased skin hydration and skin permeability of C-dots Quickly scale up production |
Poor encapsulation efficiency High water content Burst release can take place Unpredictable gelation tendency |
Kim et al. (2023) prepared a drug delivery system by encapsulating pH-sensitive C-dots and fatty acids into SLNs with minimal drug loss. The obtained SLN systems showed minimal encapsulated drugs in the blood (< 10–20%) and enhanced tumor growth inhibition efficacy [91]. Arduino et al. (2020) incorporated polyethylene glycol (PEG)-Pt(IV) prodrugs and luminescent C-dots in SLNs. The SLNs showed good stability in aqueous media and high permeability into brain tumor animal cells, suggesting the potential antitumor efficiency of SLNs [92] |
|
Hydrogels |
A large amount of water-making delivery systems is biocompatible and suitable for applications involving biological systems Slowly and steadily release C-dots over time Good biodegradable systems help to reduce environmental impacts Reduced side effects |
Sensitive to changes in temperature and pH High swelling and degradation affect the stability and functionality of hydrogels Low encapsulation efficacy High production cost |
Panda et al. (2022) developed a hydrogel system based on papaya-derived C-dots. The delivery system showed significant antibacterial activity, high drug release efficiency at B16F10 melanoma cells, and fewer side effects [93]. Turk et al. (2021) developed a novel injectable pH-sensitive C-dots and hydroxyapatite hydrogel delivery system by self-crosslinking approach. The hydrogel system showed low toxicity and a high ability to transport drugs to specific cancer cells targeted at a specifically acidic pH [94]. Wang et al. (2017) reported the photothermal-chemo therapy of chitosan C-dots loaded in a hydrogel delivery system. The hydrogel system showed high colloidal stability, loading efficiency, and stable fluorescence. The system could permeate into the tumor and release drug molecules efficiently to inhibit tumor growth [95]. |
- It is recommended that biocompatibility or toxicity analysis be discussed as a separate section, which is an important advantage of CDs to replace existing efficacy components.
Response: Thank you for your comment. The biocompatibility and toxicity of C-dots derived from plant extracts are described in section 5 “Toxicity assessment of C-dots using in cosmetic formulation”.
“5. Toxicity assessment of C-dots using in cosmetic formulation
Although the perspective properties of C-dots using in cosmetic are broad, it is also necessary to consider the potential risks due to the interactions between C-dots and biological systems. Ideally, once C-dots are penetrated in the skin layers, C-dots should perform their required functions and then be excreted from the body without any adverse effects. Therefore, it is important to comprehensively analyze the toxicity of different types of plant extract-derived C-dots from the perspective of toxicology. To assess the safety and in-vitro and in-vivo cytotoxicity, the biocompatibilities of C-dots are often determined through toxicity studies, including cell proliferation, DNA damage, oxidative stress, apoptosis, and necrosis. Skin irritation and sensitization testing (e.g., the Draize test and local lymph node assay) should be conducted to determine if C-dots can induce skin irritation and allergic reactions. From the results of these in-vitro and in-vivo evaluations, the manufactures should establish quality control measures to ensure consistent production of C-dots with minimal impurities or variation in toxicity. In addition, a regulatory compliance is required to ensure that the cosmetic product containing C-dots complies with local and international regulations for cosmetics, including safety and labeling requirements.
5.1. In-vitro toxicity
In-vitro toxicity is generally evaluated by cell proliferation during in-vitro assays (e.g., MTT, WST-1, and CCK-8 assays). Many studies have shown that cell viability negligibly decreases as adding plant extract-derived C-dots increases. Son et al. (2021) evaluated the cytotoxicity of antioxidant and anti-aging C-dots derived from tannic acid (100–500 mg/mL) using a CCK-8 assay in human melanoma skin cells (Hs 294 T) [37]. The results showed that even high C-dots concentrations (500 mg/mL) exhibited high cell viability (99.7±0.8 %) in Hs 294 T cells, suggesting low cytotoxicity. This study confirmed that C-dots are non-hazardous and can be anti-aging ingredients [37]. C-dots derived from Broccoli water extract exhibited very low cytotoxicity for A549 and 293T cells with cell viability of about 100 % and over 80 % at 100 and 500 mg/mL concentrations after 24 h of exposure, respectively [37]. The C-dots derived from carmine cochineal (Car C-dots) for lipstick applications also exhibited low toxicity even at a high concentration of about 300 mg/mL (more than 80 % HUVEC cell survival) [46]. The Car C-dots, with low toxicity and high moisture retention capability, are a potential active ingredient for lipstick application [46]. It can be seen that most plant extract-derived C-dots have good biocompatibility and low toxicity.
5.2. In-vivo
In-vivo toxicity studies involving plant extract-derived C-dots aim to evaluate their safety and potential adverse effects when introduced into living organisms. These studies include administering C-dots to animal models (e.g., mice, rats, or zebrafish embryos) and evaluating their physiological, biochemical, and behavioral effects in a specified period [71]. The biocompatibility of the Broccoli water extract-derived C-dots was evaluated in-vivo with zebrafish embryos [37]. Overall survival and hatching rates were higher than 85 %, even at the highest C-dot concentration, showing that the C-dots did not cause remarkable mortality or delayed hatching [37]. Wei et al. (2019) also evaluated the biocompatibility of Gynostemma extract-derived C-dots using in-vivo cytotoxicity on zebrafish embryos [53]. The results showed that the embryo survival rate and hatching rate were about 90 %, and the malformation rate was less than 10 %, indicating the excellent biocompatibility of the synthetic C-dots [53].
In-vivo toxicity also requires assessing the interaction of C-dots with the extracellular and the matrix of various cellular components of the tissue structure in a given organ [71]. During systemic transport, C-dots can circulate directly in the body through the blood, enter the spleen and kidney, and finally pass through the kidney filtration system to be eliminated from the body [71]. Tao et al. (2012) conducted in-vivo toxicology of C-dots (20 mg/kg) on female Balb/c mice for 90 days [72]. The careful autopsies on the 1, 7, 20, 40, and 90 days after injection of C-dots showed no organ damage and no side effects in the injected group, compared to the control group [72]. Huang et al. (2013) reported that C-dots could be rapidly and effectively eliminated from the body through urine after injection into mice [73]. Overall, the toxicity of plant extract-derived C-dots in-vitro and in-vivo is negligible, and they have good prospects for cosmetic application.”
- The author's overview of cosmetics is not comprehensive, and it is recommended that the author add whether the application of CDs in colorful cosmetics, special effects cosmetics in stage, etc., in addition to skin care products, has been reported.
Response: Thank you for your recommendation. The application of C-dots in colorful cosmetics has been described in the revised manuscript. In addition, the beneficial effects of using C-dots in skin care products are fully mentioned in section 4, “Plant extract-derived C-dots used in cosmetics”.
“4. Plant Extract-derived C-dots Used in Cosmetics
Recently, C-dots have been used in cosmetic formulations to improve the appearance of the skin due to their superior properties, including luminescent, antioxidant, anti-inflammatory and UV-absorbing properties. purple [37,40,49,64,65]. C-dots can be applied in a variety of cosmetics, such as fluorescent pigments, highlighters and illuminators, anti-UV, color-changing cosmetics, nail polish and multi-functional cosmetics. First, the luminescent properties allow the C-dots to emit bright and colorful light when exposed to specific wavelengths; therefore, they can be used as fluorescent pigments to achieve brilliant and colorful effects [65–67]. Incorporation of C-dots into cosmetics (e.g., lipstick, eye shadow, and nail polish) is expected to enhance attractive and dynamic color effects. Additionally, the C-dot's luminescent properties can create a subtle shimmer or bolder highlight, adding a colorful and glamorous dimension to makeup [64]. Therefore, plant extract-derived C-dots should be considered as the main ingredient in illuminating powders to create a bright and radiant effect on the skin. In particular, C-dots can be designed to exhibit color-changing properties based on temperature, pH, and light exposure [68]. Incorporating such color-changing C-dots into lipstick, blush, or nail polish can create products that change color based on factors such as skin pH or environmental conditions, providing a unique and customizable makeup experience [46,69]. Second, plant extract-derived C-dots extracts with anti-UV properties can also be applied as sunscreen ingredients [9,48,49]. Ideal UV absorbing and scattering properties of C-dots effectively protect the skin from harmful UV radiation while also being able to provide a color changing effect under UV light. Third, C-dots derived from plant extracts should be integrated into multifunctional cosmetic products (e.g., eyeshadows and lipsticks) to provide color and provide moisturizing, anti-aging benefits. and other skin care thanks to their versatile properties. Excellent antioxidant and anti-inflammatory properties inherited from plant extracts can help control skin aging and acne; therefore, C-dots should be included in anti-aging and acne treatment products [37,40]. It can be seen that incorporating C-dots derived from plant extracts into colorful cosmetics offers an exciting opportunity to enhance the aesthetics and functionality of makeup products, giving attractive options in the world of beauty and cosmetics. However, it is essential to ensure the safety and regulatory compliance of these products before widespread commercial use.”
- There are important references to be added on the use of carbon nanomaterials in skin care and carbon dots:Langmuir, 2019, 35, 6939-6949;ACS Appl. Bio Mater., 2020, 3, 358-368;Nanoscale, 2023, 15, 275–284;J. Mater. Chem. B, 2018, 6, 7021-7032; Separation and purification technology, 2019, 222, 60-67.
Response: Thank you for your suggestion. These references have been cited in the appropriate paragraphs.
“28. Chen, M.; Zhou, S.; Guo, L.; Wang, L.; Yao, F.; Hu, Y.; Li, H.; Hao, J. Aggregation behavior and antioxidant properties of amphiphilic fullerene C60 derivatives cofunctionalized with cationic and nonionic hydrophilic groups. Langmuir 2019, 35, 6939–6949.
- Chen, M.; Yin, K.; Zhang, G.; Liu, H.; Ning, B.; Dai, Y.; Wang, X.; Li, H.; Hao, J. Magnetic and Biocompatible Fullerenol/Fe(III) Microcapsules with Antioxidant Activities. ACS Appl. Bio Mater. 2020, 3, 358–368, doi:10.1021/acsabm.9b00857.
- Geng, Y.; Xiang, Z.; Lv, C.; Wang, Y.; Xin, X.; Yang, Y. High efficiency gold extraction through photo-luminenscent vesicles self-aggregated by sodium dodecyl sulfate and carbon quantum dots with a visual fluorescent method for Au (III) detection. Sep. Purif. Technol. 2019, 222, 60–67.
- Xu, A.; Feng, N.; Yin, K.; Li, H.; Hao, J. Supramolecular structures from structurally persistent and surface active carbon dots in water. Nanoscale 2023, 15, 275–284.
- Sun, X.; Chen, M.; Zhang, Y.; Yin, Y.; Zhang, L.; Li, H.; Hao, J. Photoluminescent and pH-responsive supramolecular structures from co-assembly of carbon quantum dots and zwitterionic surfactant micelles. J. Mater. Chem. B 2018, 6, 7021–7032.”

Round 2
Reviewer 1 Report
Dear authors
Thank you for extensively editing the manuscript. It is now in publishable form.
Best regards